# How Do Microplastics Affect Physical Properties of Silt Loam Soil under Wetting–Drying Cycles?

Xiaoyuan Jing [1], Liuchang Su [1], Yisen Wang [1], Miao Yu [2] and Xuguang Xing [1,*]

1   Key Laboratory for Agricultural Soil and Water Engineering in Arid Area of Ministry of Education, Northwest A&F University, Yangling 712100, China
2   Robinson Research Institute, Victoria University of Wellington, Wellington 5010, New Zealand
*   Correspondence: xgxing@nwsuaf.edu.cn

**Abstract:** Soil physical properties are the main factors that influence soil fertility and directly affect the soil structure and water storage capacity. Microplastics (MPs), which have caused growing concern with respect to soil pollution, have readily been detected in cultivated soils. However, the current data regarding the effects of MPs on soil physical properties during wetting–drying cycles remain insufficient. Therefore, we aimed to explore the effects of different MP particle sizes (25, 150, 550, and 1000 μm) and concentrations (1, 3, and 5%, $w/w$) on soil physical properties under indoor wetting–drying cycle conditions. The addition of MPs was found to significantly reduce the saturated hydraulic conductivity and water holding capacity of soil, while impacting the bulk density, water content, and soil particle composition. The properties of soils treated with different MP particle sizes and concentrations exhibited significant differences, while the effects of wetting–drying cycles overshadowed those of MPs. Under the wetting–drying cycles, the saturated hydraulic conductivity and initial soil water content decreased significantly, the soil water holding capacity increased, and the soil bulk density showed a trend of increasing first and then decreasing. We attribute the change to a combination of the microplastics, soil particles, and frequent wetting–drying cycles. In this type of incubation, the constant change in the soil pore proportion results in a change in water and soil porosity, and finally alters the soil physical properties. These findings demonstrate that MP accumulation, together with dynamic environmental conditions, significantly impacts the physical properties of farm land soil.

**Keywords:** microplastic; soil physical properties; wetting–drying cycles

## 1. Introduction

The accumulation of microplastics (MPs, <5 mm) in the ocean has raised increasing concern worldwide [1]. Currently, approximately 80% of marine MPs originate from soil; thus, soil represents an important source of MPs [2–4]. Although agricultural films offer diverse benefits to alter crop production by reducing water evaporation, they also represent a major source of MP pollution in soil [5]. Moreover, it has been suggested that the MP concentration in contaminated surface soil can reach 7% ($w/w$) and cause serious environmental problems [6]. Hence, considering that soil is increasingly being infiltrated by MPs, it is necessary to gain a better understanding of how MPs affect soil properties.

Soil hydraulic parameters and physical properties play key roles in root growth, as well as water and nutrient uptake, thus significantly influencing soil health [7]. Meanwhile, MP-driven changes in soil properties have emerged as possible threats to soil health. MP accumulation is consistently accompanied by changes in various soil parameters, including pore structure, bulk density, aggregate distribution, and hydraulic characteristics, which impact the soil structure integrity and alter the soil's water cycle [8–11]. In particular, de Souza Machado et al. [12] reported an increase in soil water-holding capacity and aggregate stability due to high concentrations of polyester fibers, whereas no detectable

effects were reported at a concentration of 0.1% (*w/w*) [9]. Meanwhile, Zhang et al. [11] reported that a 0.3% (*w/w*) concentration of MPs significantly decreases the number of pores at depths greater than 30 μm in clayey soils. Additionally, Guo et al. [13] reported that the infiltration capacity of soils was inversely proportional to the polypropylene concentration. Incidentally, high residual concentrations of polypropylene are commonly found in agricultural soils [14]. Wan et al. [10] also found that soil mixed with 1% MPs had the highest evaporation rate compared to treatments with microplastic levels of 0% and 0.5%. Hence, MPs pose a clear threat to the soil environment; however, there is a current dearth of data regarding the potential effects of MP particle size on soil properties, compared with the focus on the effects of MP content. Therefore, it is necessary to improve our current understanding of this aspect, in order to further explore the risks associated with different MP concentrations and sizes for soil properties.

In recent years, the increased incidence of extreme phenomena, including drought and precipitation, has led to the frequent alternation of soil drying and wetting, causing changes in soil properties [15]. Under the influence of irrigation, rainfall, and evaporation, frequent wetting–drying cycles greatly affect the soil structure and pore distribution, while also altering the movement of soil water and, thus, further impacting the hydraulic characteristics of soil, which play key roles in maintaining soil health and improving the physical quality of soil [16,17]. Relatively few studies have reported on the effect of MPs on soil parameters in relation to wetting–drying cycles. Nevertheless, one study found that wetting–drying accelerates the migration depth of MPs [18]. In addition, the combination of frequent wetting–drying cycles and MPs results in the increased formation of soil aggregates [11]. Therefore, further clarification is required regarding the addition of MPs and their potential beneficial or adverse impacts on soil properties under wetting–drying cycles.

We conducted laboratory pot-based incubation experiments with polyethylene and silt loam soil to characterize how MPs affect soil hydraulic parameters and physical properties under wetting–drying cycles. We hypothesize that MPs can alter the soil hydraulic parameters and physical properties to varying degrees depending on the wetting–drying cycles, and the wetting–drying cycle is the main influencing factor. Consequently, the main objectives of this study were to (1) investigate the relative effects of the MP concentration and particle size on soil properties; (2) elucidate the influences and importance of wetting–drying cycles, and MP concentration and particle size, on soil properties; (3) assess the relationship between wetting–drying cycles and MPs in terms of soil properties. The study is expected to provide a theoretical basis for soil environment improvement by considering the combined effects of microplastic addition and the external environment on soil.

## 2. Materials and Methods

### 2.1. Soils and MPs

The soils used in the study were silt loam obtained at a sampling depth of 30 cm and are considered to be representative of soils in Yangling, Shaanxi Province, Northwestern China (34°17′ N, 108°04′ E). The soils were air-dried, ground, and passed through a 2-mm sieve before analysis. Silt loam contains clay, silt, and sand in approximate proportions of 12.58, 72.02, and 15.40%, respectively.

Polyethylene plastic film is widely used in agricultural production. Therefore, polyethylene MPs were selected in the experiments. The experimental polyethylene MPs had a density of 0.92 g/cm$^3$. To obtain the specific microplastic particles used in our study, we purchased them from an online platform and inquired about the fabrication procedures. The method involved crushing and grinding larger plastic particles to the sizes needed for our experiments (Taiwan Formosa Plastics Company, Taiwan, China).

### 2.2. Experimental Setup

Based on reports on the common particle size and concentration of MPs in farmland [19–21], granular-shaped MP particle sizes were set to 25, 150, 550, and 1000 μm. Three MP particle concentrations were applied to the test soils (1, 3, and 5%, *w/w*). Soil without the addition of MP particles was used as the control.

The MPs were mixed with dry soil manually for 15 min and then packed into Plexiglass pots (10 cm height and 10 cm diameter) based on pre-calculated bulk densities of 1.4 g/cm$^3$. The bottom of the pot contained uniform holes to facilitate wetting. Each treatment had three replicates. Specifically, there were four microplastic levels (0%, 1%, 3%, and 5%), and four microplastic sizes (25 μm, 150 μm, 550 μm, and 1000 μm). We also conducted five wetting–drying cycles, with soil samples collected after the 0th, 1st, 3rd, and 5th cycles. The operating procedure was the same for each treatment and all measurements were performed in triplicate per treatment. Therefore, a total of 156 soil samples were included in the experiment.

The experiment was carried out in an artificial climate chamber (RXZ-400B) of the Irrigation Experimental Station of Northwest Agriculture and Forestry University. Prior to the beginning of wetting–drying cycles, all soils were saturated and maintained at saturation for 2 days to improve the initial soil water content (multiple distilled water with conductivity of approximately 1.0 μs/cm was used for saturation). The soil samples were then incubated in an artificial climate incubator (35 °C, 75RH), and the soil was remoistened to saturation after 7 days. A total of five wetting–drying alternations—9 days per wetting–drying cycle—were repeated over the course of the 60-day incubation period. Soil samples were collected at T0 (Day 0), T1 (Day 8), T3 (Day 24), and T5 (Day 40) to determine the relevant indices.

### 2.3. Measurements of Soil Physical Parameters

After sampling, the ring samples were completely saturated in distilled water for 12 h (multiple distilled water with conductivity of approximately 1.0 μs/cm was used for saturation). The soil water characteristic curve (SWRC) was then obtained by a high-speed constant-temperature refrigerated centrifuge (CR21G type II, HITACHI, Tokyo, Japan) at a constant temperature of 15 °C [22]. Finally, the van Genuchten model (Equation (1)) was used to fit the SWRCs [23].

$$\theta(h) = \frac{\theta_s - \theta_r}{\left[1 + (\alpha h)^n\right]^m} + \theta_r \tag{1}$$

where $\theta(h)$ is the soil moisture content (cm$^3$/cm$^3$); $h$ represents the soil negative pressure (cm); $\theta_s$ is the soil saturated moisture content (cm$^3$/cm$^3$); $\theta_r$ is the soil residual water content (cm$^3$/cm$^3$); $\alpha$ is the reciprocal of the air entry point (1/cm); and $n$ represents the shape parameter, $m = 1 - 1/n$ ($n > 1$).

The constant head method was used to determine the soil saturated hydraulic conductivity ($K_s$). The test was completed when the outflow was relatively unchanged between two consecutive time points [24]. The $K_s$ can be calculated by Equation (2).

$$K_s = \frac{V}{At} \cdot \frac{L}{H} \tag{2}$$

where, $V/t$ is the water flow per unit time (mL/min), $L$ and $A$ are the length (cm) and area (cm$^2$) of the soil column, respectively, and $H$ is the head pressure (cm).

The soil dry bulk density ($\gamma_d$) was determined using the dry mass of samples and the ring volume [24]. Briefly, at the end of each cycle, the blade of a 100-cm$^3$ ring knife was inserted vertically into the soil. The top cover of the ring knife containing soil samples was removed and oven-dried at 105 °C for ≥8 h until a constant weight was achieved; subsequently, the sample was weighed after cooling (accurate to 0.01 g). Soil initial water content ($\theta_i$) was computed by measuring the weight of wet soil and dry soil at the end of

each cycle. The soil particle composition was determined with an MS3000 laser particle size analyzer (wavelength: 633 nm, measurement range: 0.01–3500 micron). Approximately 0.5 g of each soil sample was air-dried and then passed through a 2-mm sieve for soil particle analysis [25,26]. The data in this study were obtained by calculating the mean value of three measurements for each sample.

### 2.4. Statistical Analysis

All statistical analyses were performed using IBM SPSS Statistics 26 and CANOCO 5. The data were screened for normal distribution using q-q plots, Shapiro–Wilk tests, and homogeneity of variance using Levene's test. One-way analysis of variance (ANOVA) with least significant difference (LSD) testing (Duncan's) was used when the data were normally distributed, and the variance of data was assumed to be equal; the effects of all three factors (concentration, size, and wetting–drying cycles), as well as their interactive effects, were analyzed using three-way ANOVA. The contribution of the treatment factors was determined through redundancy analysis by CANOCO 5. MATLAB was used to calculate the SWRC parameters with data visualization using Origin 2023.

## 3. Results

### 3.1. Effects of MP Concentrations and Sizes on Soil Hydraulic Parameters during Wetting–Drying Cycles

3.1.1. Saturated Hydraulic Conductivity

As shown in Figure 1, as the number of wetting–drying cycles increased, the $K_s$ initially decreased and then increased. At T0, $K_s$ was the highest (2.80 cm/h), while those of the other cycles were 2.61–93.63% lower compared with T0. The $K_s$ between each treatment differed significantly (Figure 1, $p < 0.05$). The most significant decrease occurred from T1 to T3 (except 1000 μm MPs). The maximum $K_s$ decrease (90.03%) occurred from T1 to T3 under a 3% treatment with 25 μm MP particles. In contrast, all treatments exhibited relatively small changes from T3 to T5, with the maximum value at either time point determined to be 0.97 cm/h. Meanwhile, under treatment with 1000 μm MP particles, the $K_s$ decreased sharply by T1 (84.01–91.31%) compared to T0, and then remained relatively stable (Figure 1d).

$K_s$ was also affected by the concentration of MPs (Figure 1); however, this effect varied among the different wetting–drying cycles and based on the particle size. The addition of MPs significantly decreased the $K_s$ by 1.25–46.43% compared to the CK treatment at T0. More specifically, at T0, the $K_s$ exhibited a decreasing trend with increased MP concentration at all sizes. However, treatment with 1000 μm MP particles showed the opposite trend; that is, $K_s$ increased with increasing MP concentration. Changes in MP concentrations at all other time points (T1, T3, T5) accounted for 12.59–123.92, 82.2–228.44, and 63.99–242.33% changes in $K_s$ compared to the CK treatment, respectively.

Size-wise comparisons showed that the MP particle size also altered the $K_s$ values of the soil, with significant differences observed between treatments with different sizes (Table S1; $p < 0.05$).

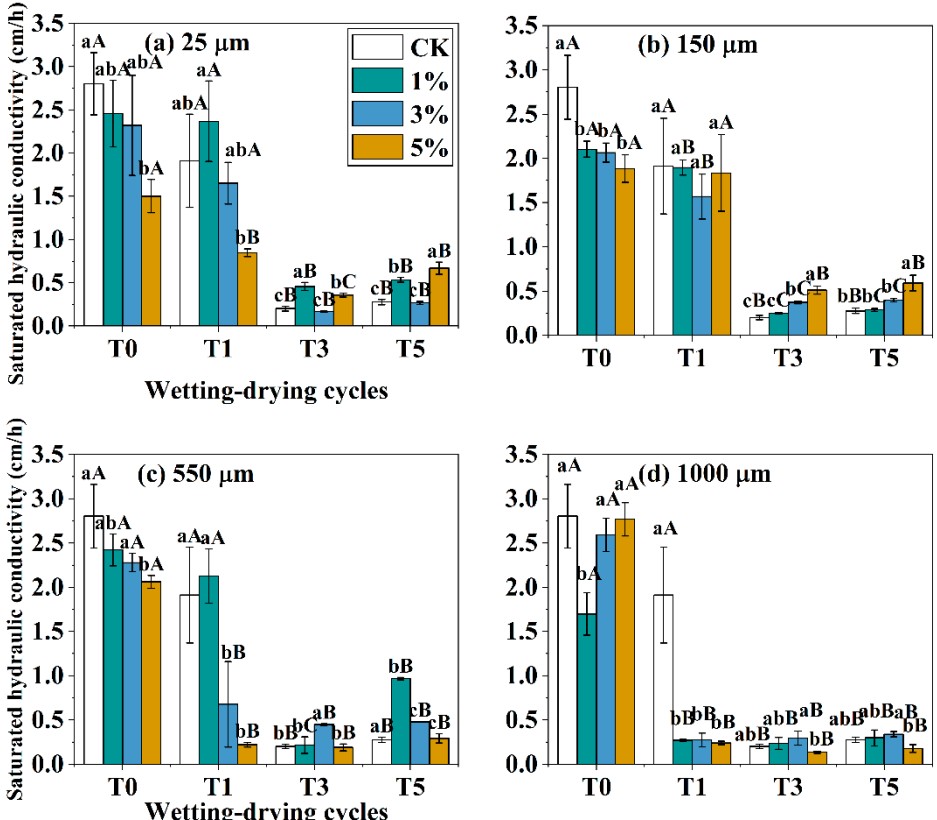

**Figure 1.** Effects of microplastic particle sizes (**a**) 25 μm, (**b**) 150 μm, (**c**) 550 μm, and (**d**) 1000 μm during wetting–drying cycles on soil saturated hydraulic conductivity. The lowercase letters a, b, and c represent significant differences between treatments with the same MP particle size and different concentrations. The uppercase letters A, B, and C represent significant differences between treatments with the same concentration and different wetting–drying cycles ($p < 0.05$). CK (white) represents 0% microplastic addition level, 1% (green) represents 1% microplastic addition level, 3% (blue) represents 3% microplastic addition level, and 5% (dark yellow) represents 5% microplastic addition level. T0, T1, T3, and T5 represent the zero, first, third, and fifth wetting–drying cycles.

3.1.2. Soil Water Retention Curve (SWRC)

Figure 2 highlights the SWRC of soil by considering different wetting–drying cycles. It was not observed that the dynamic characteristics of the soil volumetric water content differed significantly among treatments with a soil suction < 1000 cm (Figure 2). The difference in volumetric water content is reflected by a higher suction range (1000–7000 cm). In this range, the average water content of T1, T3, T5 with MPs was greater than that at T0 (Figure 2a–d). For example, compared to T0, the average SWRC increased by 0.22–13.04, 0.57–13.21, 3.07–16.40, and 5.24–15.64%, respectively, in soil treated with 25, 150, 550, and 1000 μm MP particles, from T1 to T3. However, these changes were lower than that observed for the SWRCs under the CK treatment (Figure 2e), the average water content for which reached a maximum value of 0.31 cm$^3$/cm$^3$, at T5. Based on the fitting parameters (listed in Table S2), a noticeable change in $\theta_s$ and $\theta_r$ was observed between different treatments during the wetting–drying stages. That is, within the same treatment group, higher $\theta_s$ was observed before wetting–drying cycles and subsequent decreasing trends were observed with an increase in the number of wetting–drying cycles. The minimum $\theta_r$ values were detected at T1 for all treatments, excluding the treatment with 1% of 25 μm MP. Collectively, these results indicate that the experimental conditions impacted the SWRCs. Specifically, treatment with 25 μm MP particles exhibited the greatest change at T3, while this was observed for the other treatments at T5.

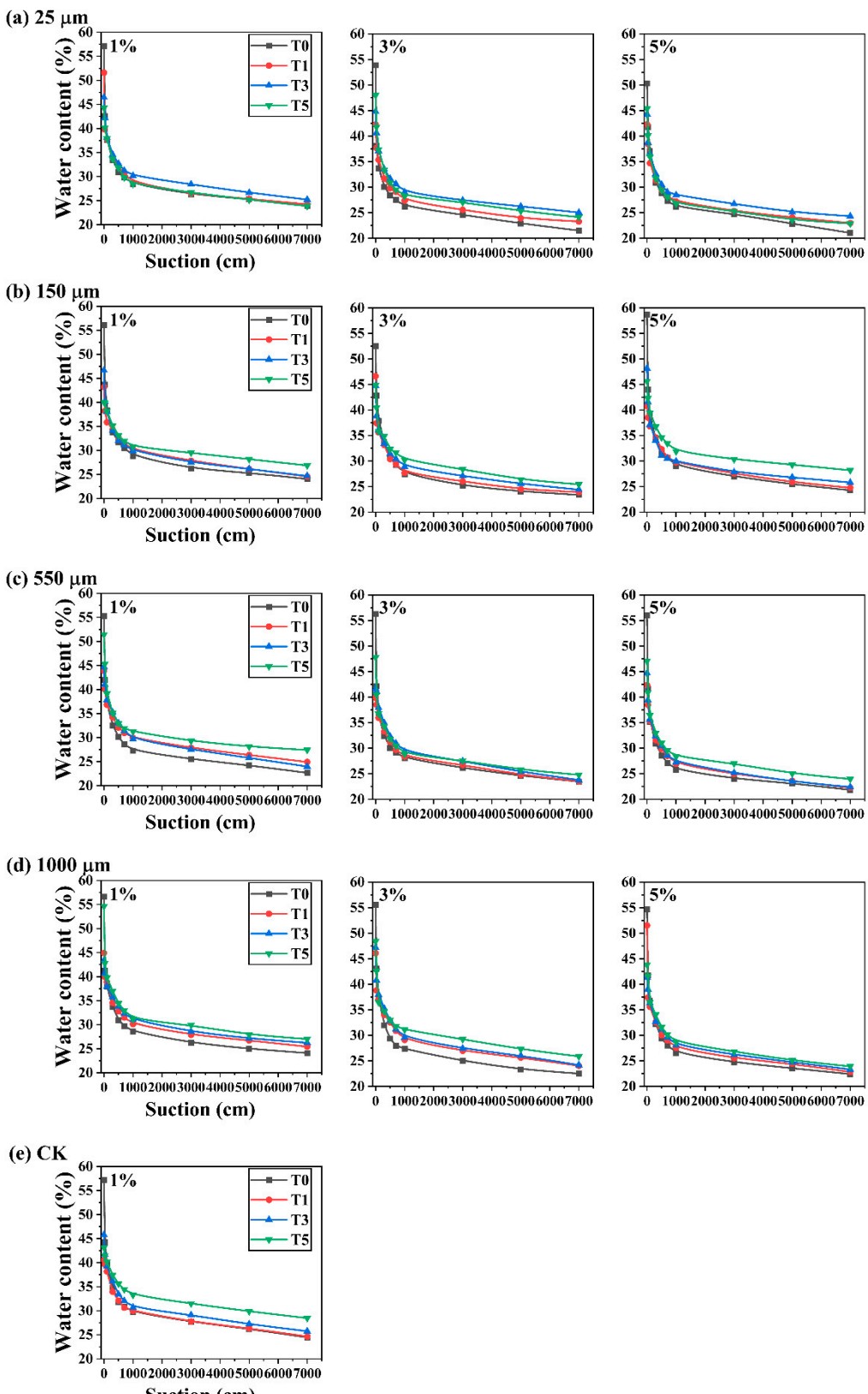

**Figure 2.** Effect of treating soil with the same concentration and different sizes of microplastics (**a**) 25 μm, (**b**) 150 μm, (**c**) 550 μm, and (**d**) 1000 μm, (**e**) CK, under different wetting–drying cycles on the soil water retention curves. T0 (black) represents the 0th wetting–drying cycle, T1 (red) represents the first wetting–drying cycle, T3 (blue) represents the third wetting–drying cycle, and T5 (green) represents the fifth wetting–drying cycle.

Additionally, the MP concentration also influenced the SWRCs (Figure 3). With an increase in the MP concentration, soil water content levels decreased in the following order CK > 1% > 3% > 5%. However, under treatment with 150 μm MP particles, the lower SWRCs were observed at a 3% concentration. When the MP concentration was 5%, the decrease in water content was relatively greater than those in other treatment groups, particularly under the 25 μm treatment, which retained 20% less water than did the CK treatment. In addition, the $\theta_s$ showed an overall decreasing trend with increasing MP concentrations at T0 (except for the 150 μm treatment). During the subsequent cycle, the value of $\theta_s$ increased and then decreased only with an increasing concentration in the 150 μm treatment, while no consistent changes were observed in the other treatments (Table S2). Meanwhile, no consistent relationship was observed for $\theta_r$ variation between each wetting–drying cycle.

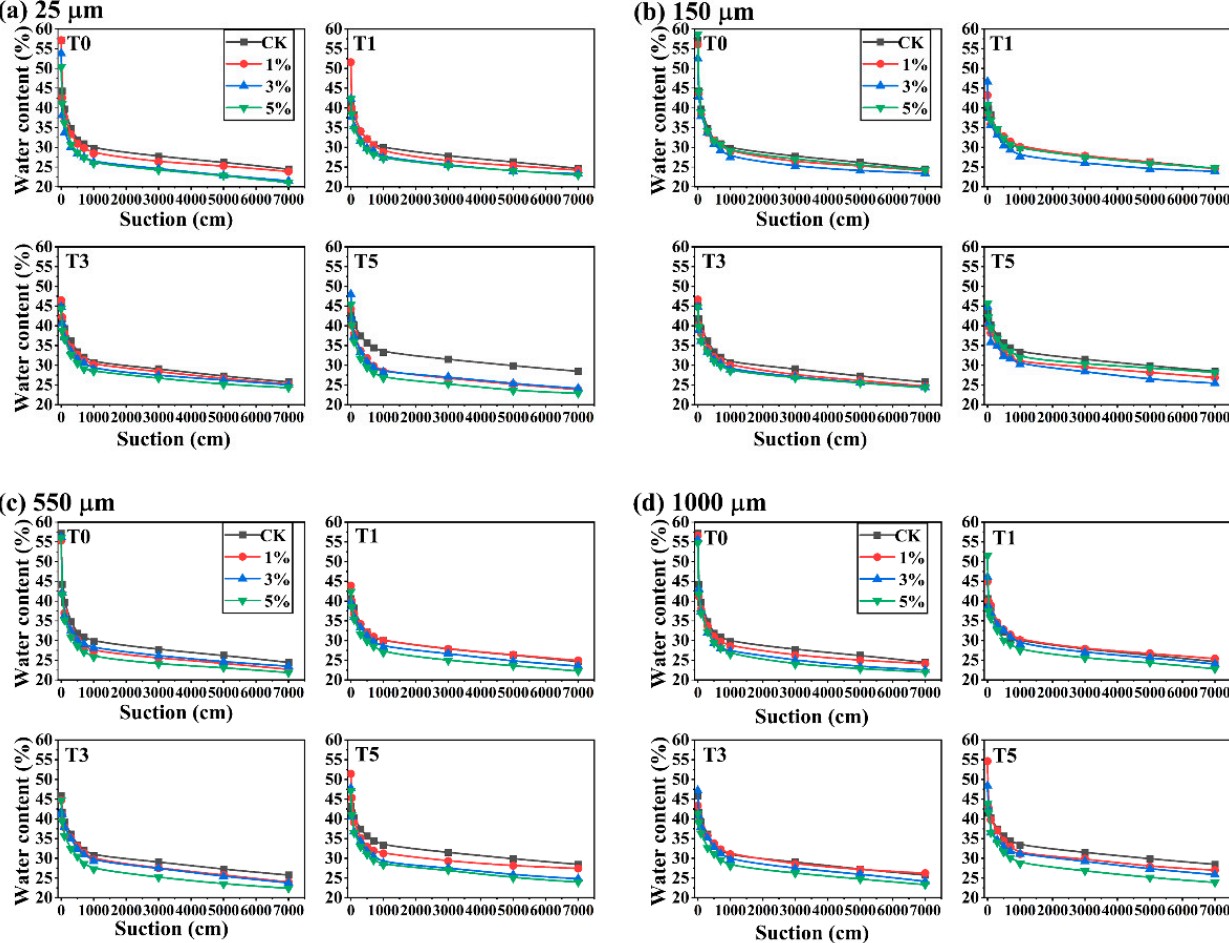

**Figure 3.** Effect of treating soil with different concentrations and sizes of microplastics (**a**) 25 μm, (**b**) 150 μm, (**c**) 550 μm, and (**d**) 1000 μm under the same wetting–drying cycles on the soil water retention curves. CK (black) represents 0% microplastic addition level, 1% (red) represents 1% microplastic addition level, 3% (blue) represents 3% microplastic addition level, and 5% (green) represents 5% microplastic addition level.

Although the MP particle size also affected the SWRCs (Figure 4), the effects appeared to be inconsistent in different wetting–drying cycles. While no distinct changes were observed in soil treated with low MP concentrations during the first two cycles (T0 and T1), the average soil water content in samples treated with 25–1000 μm MP particles decreased, respectively, by 13.01, 2.21, 12.31, and 11.87%, at 5% MP application, compared with the CK treatment before the wetting–drying cycles began (T0). Thereafter, minimal changes

occurred in the average soil water content of samples treated with 150 μm MPs, with only a 0.01% decrease compared to the CK treatment at T1. Notably, in samples treated with 25 μm MPs, the average soil water content was consistently the lowest at the end of the wetting–drying cycles (T5). Moreover, only at T0, the $\theta_s$ was found to decrease with an increase in MP particle size (Table S2); in contrast, the $\theta_r$ values fluctuated.

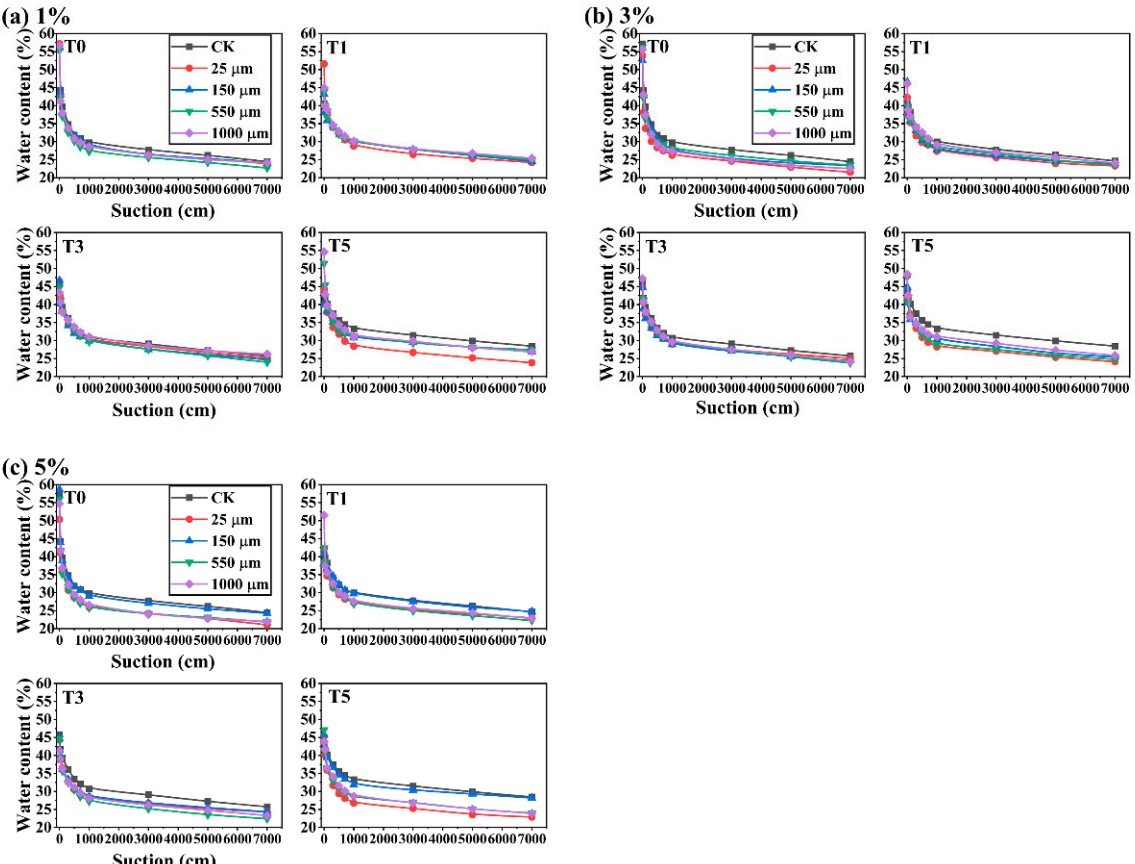

**Figure 4.** Effect of treating soil with differentsizes and the same concentration of microplastics (**a**) 1%, (**b**) 3%, (**c**) 5% under the same wetting–drying cycles on the soil water retention curves. CK (black) represents 0% microplastic addition level, 25 μm (red) represents microplastics with a particle size of 25 μm, 150 μm (blue) represents microplastics with a particle size of 150 μm, 550 μm (green) represents microplastics with a particle size of 550 μm, 1000 μm (purple) represents microplastics with a particle size of 1000 μm.

### 3.1.3. Bulk Density

Treatment of soil with MPs resulted in an altered initial $\gamma_d$. Meanwhile, the wetting–drying cycles also played a crucial role (Figure 5). After two wetting–drying cycles (T1 and T3), a rapid increase, from 1.4 to 1.60 g/cm³, was observed in the $\gamma_d$ of soil treated with 150 μm, 1% MP particles (Figure 5b). Compared to an initial bulk density of 1.4 g/cm³, the soil bulk density increased by 1.87–14.89% after T3 for all treatments with added microplastics. Thereafter, significantly lower bulk densities were observed at T5 compared to T3; however, they remained higher than those at T0 (except for 1% treatment with 25 μm MP particles). Furthermore, the soil bulk density of the CK treatment progressively increased with the wetting–drying cycles (1.42, 1.48, 1.60 g/cm³; $p < 0.05$; Figure 5). This change was also reflected in the application of 5% 1000 μm MP particles (1.45, 1.46, and 1.50 g/cm³; $p < 0.05$; Figure 5d). In addition, slight differences were observed in the soil bulk density between the CK treatment and the MP treatment during the wetting–drying cycles ($p < 0.05$; Figure 4), as well as between treatments with different particle sizes ($p < 0.05$; Figure 5, Table S3).

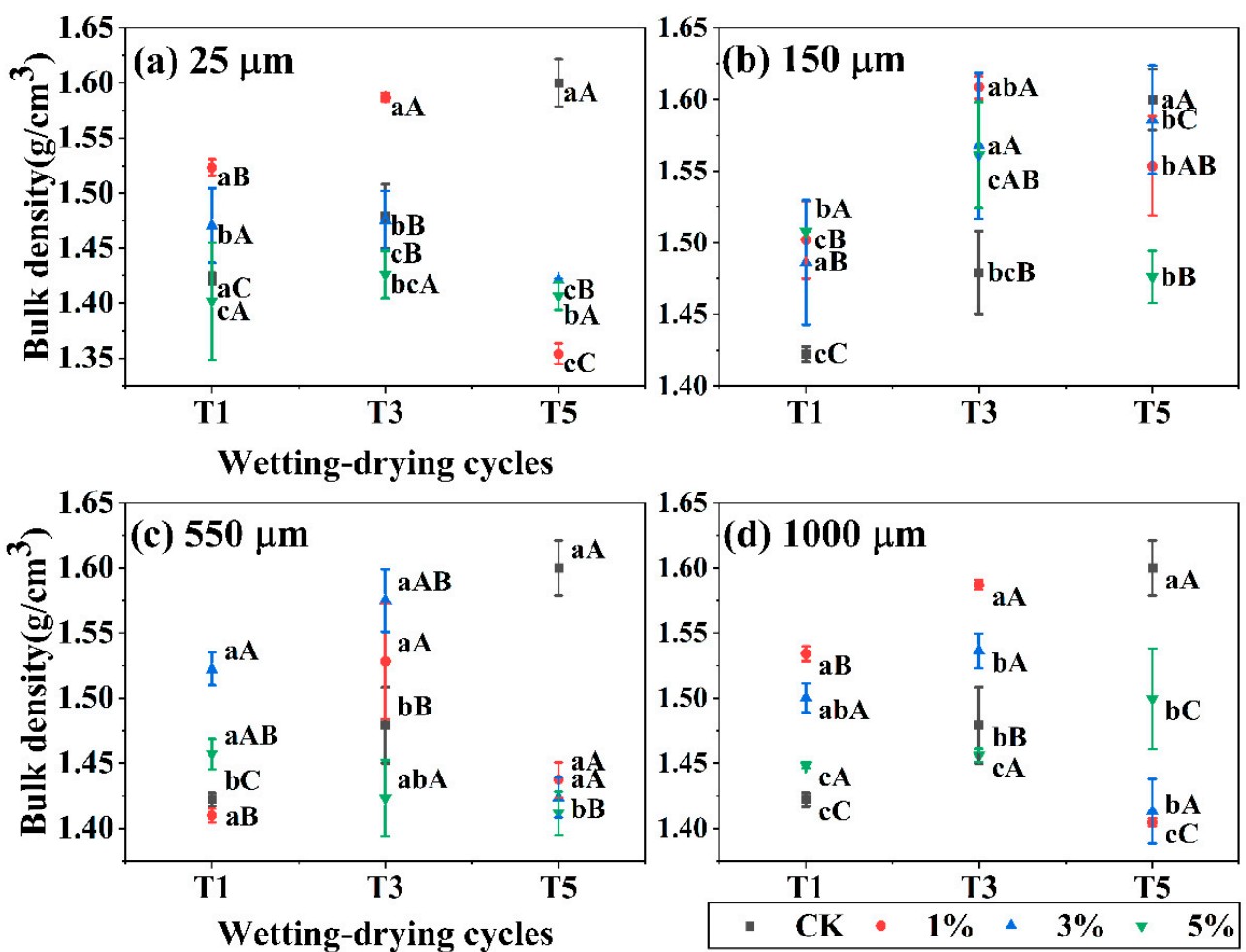

**Figure 5.** Effect of treating soil with different sizes of microplastics (**a**) 25 μm, (**b**) 150 μm, (**c**) 550 μm, and (**d**) 1000 μm on soil bulk density. The lowercase letters a, b, and c represent significant differences between treatments with the same microplastic particle size and different concentrations; the uppercase letters A, B, and C represent significant differences between treatments with microplastics and different wetting–drying cycles ($p < 0.05$). CK (black) represents 0% microplastic addition level, 1% (red) represents 1% microplastic addition level, 3% (blue) represents 3% microplastic addition level, and 5% (green) represents 5% microplastic addition level.

### 3.1.4. Water Content

Soil water content was impacted differently by MP treatments and wetting–drying cycles (Figure 6). An overall decreasing trend was observed in all treatments with an increase in wetting–drying cycles, even if slight increases were observed at T3, as was observed for the 150 μm-3%, 550 μm-1%, and 1000 μm-1% treatments (Figure 6b–d). The relatively lowest values of 0.15, 0.14, and 0.17 $cm^3 \cdot cm^{-3}$ occurred under treatments with 1, 3, and 5% of 25 μm MP particles at T5, respectively. However, no significant differences were observed between the CK treatment and MP application level ($p > 0.05$; Figure 6), while different particle sizes caused significant differences (except 3% and 5% treatment at T1; $p < 0.05$; Table S3).

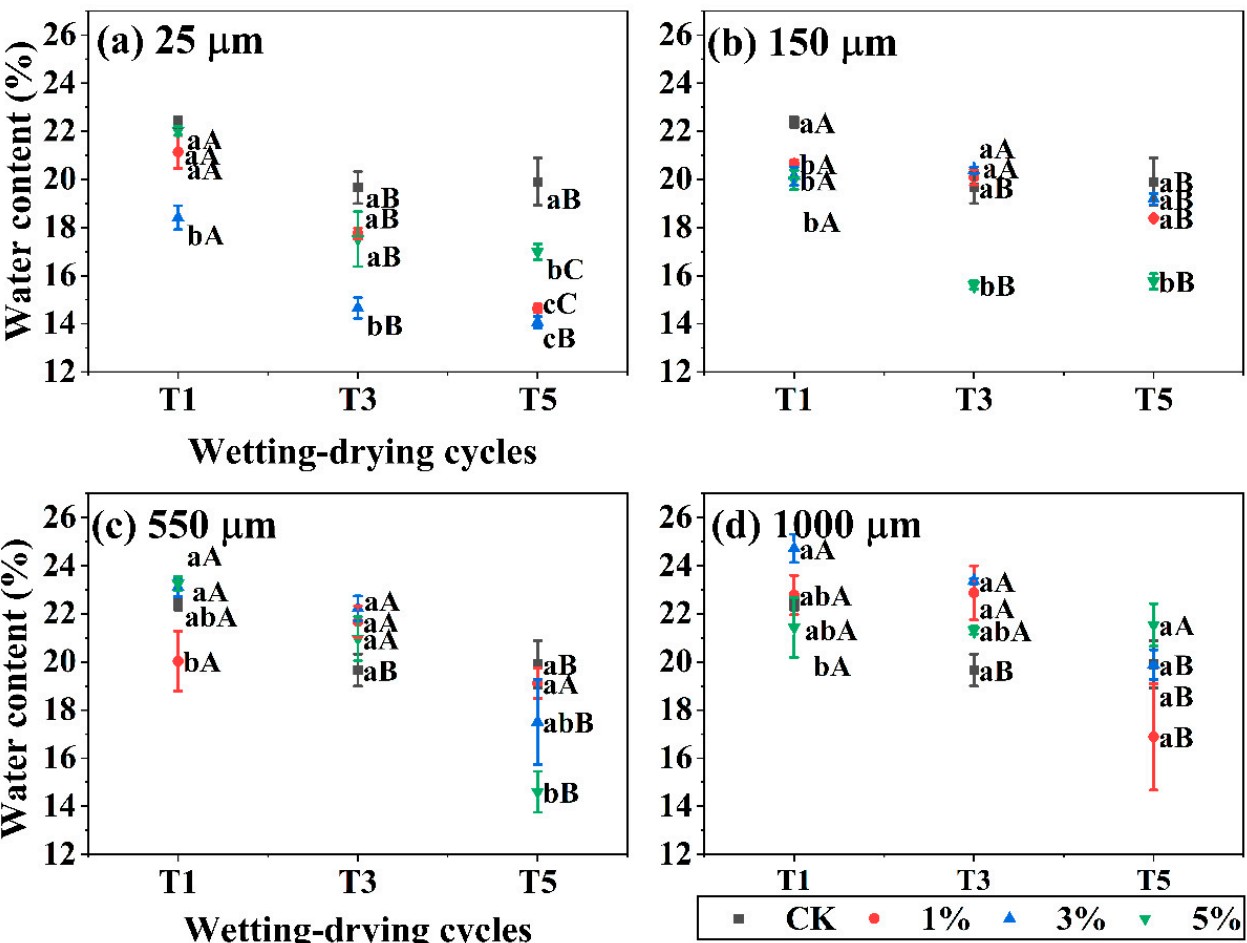

**Figure 6.** Effect of treating soil with different sizes of microplastics (**a**) 25 μm, (**b**) 150 μm, (**c**) 550 μm, and (**d**) 1000 μm on initial soil water content. The lowercase letters a, b, and c represent significant differences between treatments with microplastics of the same particle size and different concentrations; the uppercase letters A, B, and C represent significant differences between treatments with microplastics and different wetting–drying cycles ($p < 0.05$). CK (black) represents 0% microplastic addition level, 1% (red) represents 1% microplastic addition level, 3% (blue) represents 3% microplastic addition level, and 5% (green) represents 5% microplastic addition level.

### 3.1.5. Soil Particle Composition

Analysis of the soil particle composition revealed no significant ($p > 0.05$) changes in the samples treated with different concentrations of MPs. However, the proportion of soil silt increased following all treatments with MPs compared to the soil initial silt content ($p < 0.05$, Figure 7). The soil particle composition was affected by the MP particle size ($p < 0.05$). The proportion of silt particles in soil increased when the microplastic particle size increased from 150 μm to 1000 μm.

### 3.2. Joint Influence Mechanism

#### 3.2.1. Interactive Effects of MP Concentration and Particle Size and Wetting–Drying Cycles on Soil Parameters

Wetting–drying cycles had significant effects on most soil parameters except the soil particle composition, and accounted for 67.98, 32.35, 82.03, 67.21, and 15.14% of the variability in the $K_s$, $\gamma_d$, $\theta_i$, $\theta_s$, and $\theta_r$, respectively (Table 1). The MP concentration and particle size also significantly impacted all the soil parameters; however, they accounted for a relatively small proportion of the variability—for example, concentration and particle size accounted for 0.01 and 0.08% of the variability in the initial water content (Table 1).

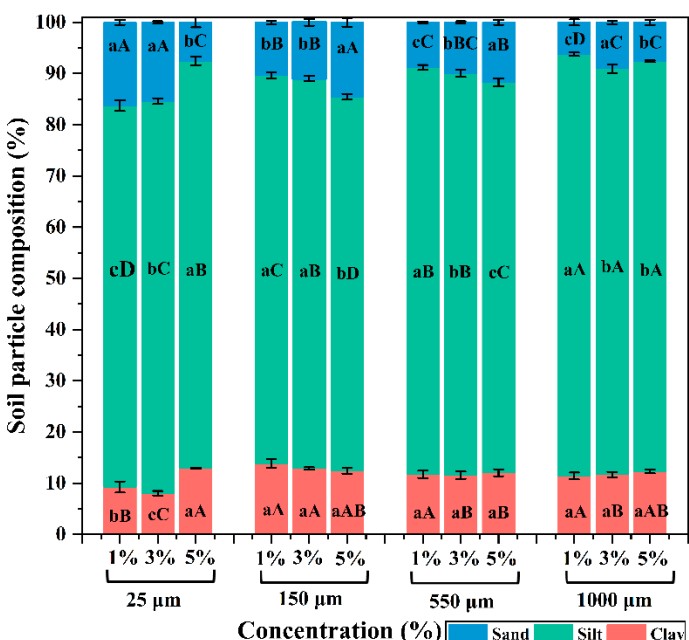

**Figure 7.** Effect of treating soil with microplastics on soil particle composition after wetting–drying cycles. The lowercase letters a, b, and c represent significant differences between samples treated with the same microplastic particle size and different concentrations; the uppercase letters A, B, and C represent significant differences between samples treated with the same microplastic concentration and different particle sizes ($p < 0.05$). Sand (blue) represents sand proportion in the test soil, Silt (green) represents silt proportion in the test soil, and Clay (red) represents clay proportion in the test soil. The horizontal axis indicates that the microplastics with particle sizes of 25 μm, 150 μm, 550 μm, and 1000 μm microplastics had three additive levels of 1%, 3%, and 5%, respectively.

**Table 1.** Contributions made by independent factors (wetting–drying cycles, concentration, and particle size of microplastics) and their combined interactions to soil parameters studied by three-way ANOVA.

| Parameters | Wetting–Drying Cycles | | Concentration | | Size | | Wetting–Drying Cycles × MP Concentration | | Wetting–Drying Cycles × MP Size | | Concentration × Size | | Wetting–Drying Cycles × MP Concentration × MP Size | | Residual |
|---|---|---|---|---|---|---|---|---|---|---|---|---|---|---|---|
| | % | sig. | % | sig. | % | sig. | % | sig. | % | sig. | % | sig. | % | sig. | |
| $K_s$ | 67.98 | 0.000 | 1.40 | 0.000 | 2.49 | 0.000 | 2.74 | 0.000 | 8.53 | 0.000 | 3.37 | 0.000 | 4.10 | 0.000 | 9.49 |
| $\gamma_d$ | 32.35 | 0.000 | 3.66 | 0.000 | 11.11 | 0.000 | 7.45 | 0.000 | 7.31 | 0.000 | 2.81 | 0.000 | 10.41 | 0.000 | 24.89 |
| $\theta_i$ | 82.03 | 0.000 | 0.01 | 0.036 | 0.08 | 0.000 | 0.02 | 0.015 | 0.06 | 0.000 | 0.04 | 0.000 | 0.08 | 0.000 | 17.68 |
| $\theta_s$ | 67.21 | 0.000 | 1.43 | 0.003 | 1.43 | 0.008 | 0.85 | 0.272 | 5.62 | 0.000 | 1.96 | 0.013 | 8.09 | 0.000 | 13.41 |
| $\theta_r$ | 15.14 | 0.000 | 3.92 | 0.041 | 2.56 | 0.230 | 7.04 | 0.077 | 19.27 | 0.001 | 3.83 | 0.371 | 15.05 | 0.148 | 33.19 |
| Clay | | | 12.43 | 0.000 | 38.68 | 0.000 | | | | | 36.68 | 0.000 | | | 12.21 |
| Silt | | | 1.96 | 0.000 | 49.19 | 0.000 | | | | | 36.31 | 0.000 | | | 12.54 |
| Sand | | | 1.35 | 0.000 | 39.77 | 0.000 | | | | | 51.83 | 0.000 | | | 7.06 |

Note: $K_s$ represents soil saturated hydraulic conductivity. $\gamma_d$ represents soil bulk density. $\theta_i$ represents initial soil water content. $\theta_s$ represents saturated water content. $\theta_r$ represents residual water content. Sand represents the proportion of sand in soil. Silt represents the proportion of silt in soil. Clay represents the proportion of clay in soil.

The combined effect of wetting–drying cycles and MP concentration was significant on $K_s$, $\gamma_d$, and $\theta_i$, but not on $\theta_s$ or $\theta_r$ (Table 1). $K_s$, $\gamma_d$, $\theta_i$, $\theta_s$, and $\theta_r$ were also significantly affected by the combined effect of wetting–drying cycles and MP size, while nearly all soil parameters, excluding $\theta_r$, were significantly affected by the combined effect of MP concentration and MP size. In addition, the combined effect of wetting–drying cycles, MP concentration, and MP particle size significantly impacted all soil parameters except soil particle composition and $\theta_r$. These three factors and their interactions explained <50% of the variability according to their residual contributions to these parameters.

### 3.2.2. Relationships between Experimental Factors and Soil Parameters

The relationships among the measured parameters of soil and treatment factors (wetting–drying cycles, concentration and particle size of MPs) were depicted in a correlation heat map and ordination diagram (Figure 8). The Monte Carlo permutation tests indicated significant differences among all canonical axes ($p < 0.01$); the first axis explained 27.42% of the variation in the parameter–factor relationships (Figure 8a; Table S4). Specifically, the wetting–drying cycles had a larger negative effect on $\theta_i$ and $K_s$, followed by $\theta_s$ ($p < 0.05$, Figure 8b). However, the wetting–drying cycles had a smaller positive effect on $\gamma_d$ and $\theta_r$. The MP concentration had a smaller negative effect on $K_s$, $\gamma_d$, $\theta_s$, and $\theta_i$, and a positive effect on $\theta_r$. The MP particle size had a negative effect on $K_s$ and $\gamma_d$, and a slight positive effect on $\theta_r$, $\theta_s$, and $\theta_i$.

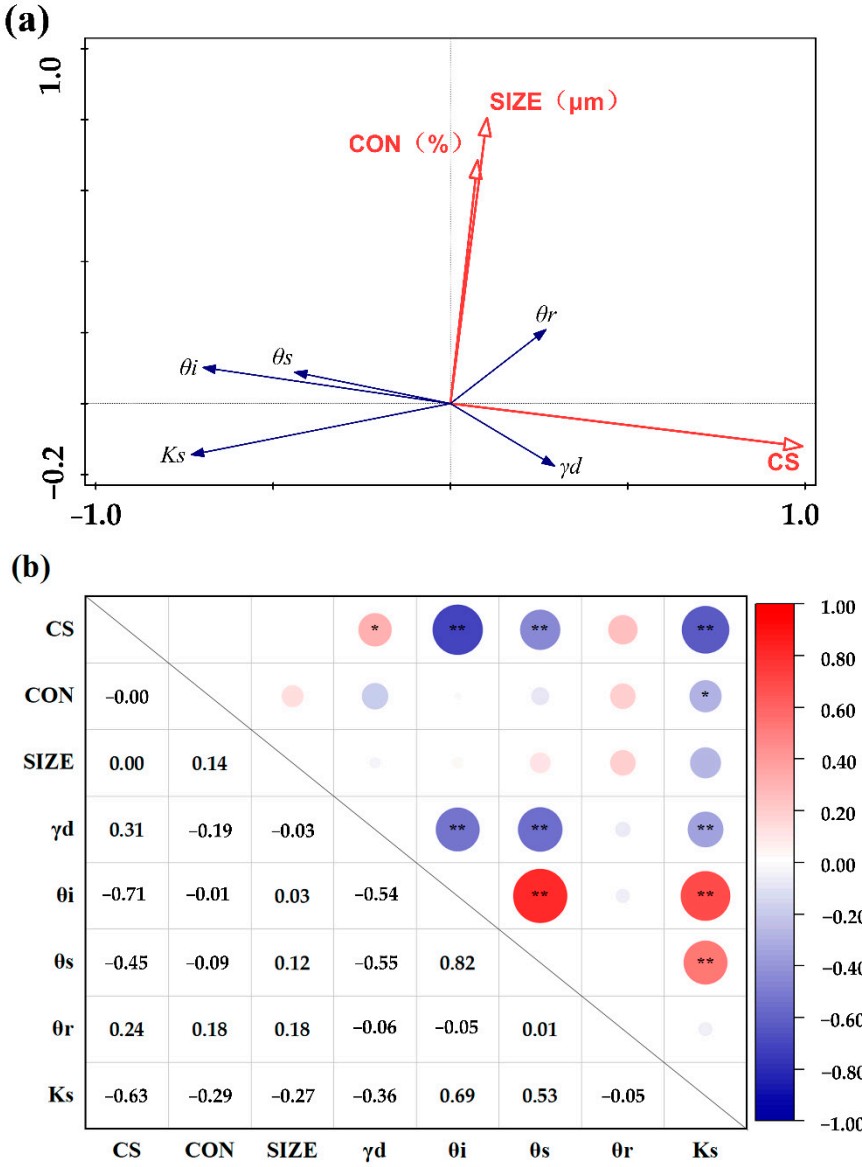

**Figure 8.** Associations between soil physical properties and environmental factors. (**a**) Redundancy analysis ordination diagram of soil parameters with treatment factors. Environmental factors are represented by red arrows. Soil properties are indicated by blue arrows. (**b**) Pearson's rank correlations

of soil physical properties with wetting–drying cycles, microplastic particle size, and concentration. Red represents positive correlations, blue represents negative correlations. CS represents wetting–drying cycles (T0, T1, T3, and T5). CON represents microplastic concentration (CK, 1, 3, and 5%). SIZE represents microplastic particle size (CK, 25, 150, 550, and 1000 μm). $\theta_i$ represents initial soil water content. $K_s$ represents soil saturated hydraulic conductivity. $\theta_s$ represents saturated water content. $\theta_r$ represents residual water content. $\gamma_d$ represents soil bulk density.

## 4. Discussion

This study shows that different concentrations and sizes of MPs impact soil physical properties in different ways. The wetting–drying cycles significantly altered the soil physical properties and overshadowed the effects elicited by the MPs. More specifically, under the influence of repeated wetting–drying cycles, the soil physical properties exhibited strong temporal changes. Moreover, MP-driven changes in soil properties were highly dependent on wetting–drying cycles.

$K_s$ can characterize the soil water infiltration capacity [27]. Herein, the water infiltration capacity of the soils was found to be inversely proportional to the MP concentration before the wetting–drying cycles (T0). This supports the findings of Guo et al. [13], who reported that an increase in MP concentration alters the soil permeability by occupying the original soil pores. This concept is also supported by previous studies that reported on the ability of MP particles to occupy the original soil pores and increase the proportion of large pores, which is not conducive to water movement [28,29]. Moreover, the addition of MPs causes the specific surface area and porosity of soil to decrease, which directly determines the soil water conservation level [30]. This likely accounts for the observed decrease in the soil water holding capacity in the current study. Moreover, this adverse effect is attributed to the strong hydrophobicity of MPs [31]. Indeed, an increase in hydrophobicity negatively impacts soil hydraulic properties [32].

Additionally, the MP particle size may affect the pore space and particle interactions within the soil. That is, small particles may affect soil micropores, causing an increase in the water retention capacity of soil by increasing the specific surface area [33,34]. However, this contradicts the findings of the current study. This discrepancy may be due to the different soil textures and hydrophobicity caused by the MPs [31,35,36]. Moreover, small MP particles can reside in the macropores, thus reducing the number of macropores, restricting channels of water movement, and reducing the water infiltration capacity of soil [13,37]. In contrast, large particles mixed with soil occupy the small and medium pore spaces in the soil, causing an increase in the volume of large pores, resulting in an increase in hydraulic conductivity and further accelerated water infiltration [30]. This phenomenon was partially observed in our study (Table S1). Although the different particle sizes caused changes, linear changes in parameters were only observed at higher MP concentrations. Hence, some of the small MPs appeared to induce stronger effects than did the larger MPs, compared with the control. Our findings suggest that the application of MPs impacts soil particle composition, soil bulk density, and initial water content; however, the effect is not directly proportional to the concentration or particle size of the MP. This is likely due to the influence of the wetting–drying cycles. That is, once the MP-treated soil undergoes repeated wetting–drying cycles, changes caused by the MP concentration or particle size become secondary to those caused by the wetting–drying cycles. Although the properties of soils treated with different MP particle sizes and concentrations exhibited significant differences, more notable effects were observed under the influence of all three factors combined. Thus, the wetting–drying cycles appear to have a leading role in the coupling of these three factors (Table 1; Figure 8).

From water absorption to water loss, the pressure exerted by water molecules on the soil mixtures was noticeable after the first wetting–drying cycle, causing the compaction of the entire soil space, resulting in increased water infiltration resistance and soil bulk density [38,39]. Subsequently, frequent wetting–drying cycles weaken the compaction effect of water on the soil mixture. Moreover, the hydrophobicity of MPs strongly repels water

molecules, thus increasing particle aggregation and decreasing the specific surface area of the soil. This may result in the generation of a new channel to facilitate water movement, leading to an increase in water infiltration capacity [40]. Similarly, the water causes the particles in the "soil–MP" mixture to aggregate and enter the macropores, causing a reduction in macropores and total porosity. Consequently, the number of micropores and mixture particles increases, thereby increasing the water holding capacity [41]. This concept is also supported by Bodner et al. [41], who suggested that the increase in small pores during the drying stage leads to an increase in soil water holding capacity. Hence, the positive effects of the wetting–drying cycles on soil water holding capacity exceeded the negative effects of the MPs; however, this does not negate the adverse effects of the MP concentration (Figure 3).

Overall, these results indicate that the combination of wetting–drying cycles and MP application alters the soil porosity and pore size distribution, increasing soil water retention. The resulting rearrangement of soil particles and pore size distribution have significant impacts on the hydraulic conductivity and water retention properties [42]. Additionally, the reduction in bulk density and $K_s$ is attributed to the dense packing of soil particles and increased microporosity, resulting from wetting–drying cycles [33].

However, studies have shown that the addition of MPs to soil decreases the saturated hydraulic conductivity and water holding capacity, and can impact soil bulk density. Once incorporated into soil, MPs can be retained for many years, potentially posing a significant long-term threat to soil quality, soil microorganism composition, soil animals, and plants [43–47]. Similarly, soil that is subjected to long-term wetting–drying cycles is likely to exhibit changes in fertility before tillage, and wetting–drying cycles are able to impact the agricultural ecosystem via alteration of the microbial community and related behaviors [48–51]. Therefore, our results show that the negative effects of wetting–drying cycles on soil under the conditions of MP storage must also be carefully considered. However, the influence and duration of effects elicited by wetting–drying cycles on soil properties following the addition of MPs, as well as the unpredictable interactions among them, require more in-depth investigation.

## 5. Conclusions

The results of this study indicate that MPs affect the soil saturated hydraulic conductivity, water holding capacity, soil bulk density, initial soil water content, and soil particle size distribution. Moreover, the application of wetting–drying cycles alters the soil moisture and pore distribution, and plays a major role in influencing soil properties. Collectively, these findings support the need for reduced MP pollution in farmland soil and provide a theoretical basis for the water-saving irrigation of farmland soil, thus laying a foundation for future soil moisture research and sustainable agricultural development. It should be noted that other factors, which were not considered in this study, can also affect the physical properties of soil. In addition, the simulation control of indoor environmental factors does not entirely reflect environmental changes under natural conditions. Moreover, the concentration of microplastics in the actual farmland environment may be lower than the conditions applied in this experiment. Therefore, based on our current understanding, it is necessary to strengthen research on the actual soil conditions in farmland, and further research is also required to fully understand the effects of MPs and external conditions, such as wetting–drying cycles, on soil and the agroecosystem. Overall, this case study demonstrates that the accumulation of microplastics and dynamic environmental conditions will affect farmland soil. Consideration under the external conditions experienced by the soil is required to assess the long-term effects of microplastics on soil properties.

**Supplementary Materials:** The following supporting information can be downloaded at: https://www.mdpi.com/article/10.3390/agronomy13030844/s1, Table S1: Effects of microplastics with the same concentration and wet-dry cycles and different sizes on soil saturated hydraulic conductivity ($K_s$); Table S2: Van-Genuchten (V-G) model fitting parameters; Table S3: Effects of

microplastics with the same concentration and wet-dry cycles and different sizes on soil bulk density ($\gamma_d$) and initial moisture content ($\theta_i$); Table S4: RDA summary.

**Author Contributions:** Conceptualization, data analysis, writing—original draft, X.J.; investigation, formal analysis, methodology, data curation, X.J. and L.S.; supervision and visualization, Y.W.; data curation and writing, M.Y.; writing—review and editing, X.X. and M.Y.; funding acquisition and project administration, X.X. All authors have read and agreed to the published version of the manuscript.

**Funding:** This work was financially supported by the Innovation and Entrepreneurship Training Program for College Students—Innovation Training Program (S202210712314) and the National Natural Science Foundation of China (51809217).

**Data Availability Statement:** Data will be made available on request.

**Acknowledgments:** This work was financially supported by the Innovation and Entrepreneurship Training Program for College Students—Innovation Training Program (S202210712314) and the National Natural Science Foundation of China (51809217).

**Conflicts of Interest:** The authors declare that they have no known competing financial interest or personal relationships that could have appeared to influence the work reported in this paper.

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
