# Peer review of "How Do Microplastics Affect Physical Properties of Silt Loam Soil under Wetting–Drying Cycles?"

_agronomy, doi:10.3390/agronomy13030844_

Round 1

Reviewer 2 Report

In this manuscript, the authors aimed to explore the effects of different MP particle sizes and concentrations on soil physical properties under indoor wetting–drying cycles conditions. This manuscript presented solid data, but improvement has to be made to warrant its acceptance to this journal. I have the following concerns:

1.      P2 L88, explain “MPs were prepared from polyethylene for agriculture”

2.      P2 L92, how did the authors prepared granular shaped MP particles, and I think there is impossible to prepare particles at the exact size, like 25, 250 … It should be a size range if you used the method of sieving.

3.      P3 L97-99, “each treatment performed in triplicate” repeated twice?

4.      P14 L65, “the wetting–drying cycles appear to have a leading role in the 65 coupling of these three factors”, and together with the results in the whole manuscript, it seems that the effect of MPs was minor. And the concentrations of MPs used were actually higher than that in real environment, so the authors may need to attenuate your statements somewhat to indicate that this is more of a mechanistic paper rather that an indication of what is likely to occur in the environment under current conditions.  This could especially be objectively addressed in the Conclusions section.

Reviewer 3 Report

Thank you for the possibility to review the paper submitted by Jing et al. The paper focuses on the influence of microplastic concentration and particle size on the soil's physical properties (water storage capacity, infiltration, density) under wetting-drying cycles in laboratory conditions. The topic is very interesting and still needs to be more recognized. The paper is very clear and well-organized. Data are shown in Figures and Tables sufficiently and clearly, and the introduction and Discussion sections have appropriate references. In general, I do not have any comments or suggestions for this paper, and I accept it for publication in its present form. 

Author Response

Dear Editors and Reviewers

Thank you for taking the time to review my manuscript.  Your feedback and support are greatly appreciated. I am grateful for your efforts in reviewing the paper, even though you did not provide any specific comments or suggestions.

Your time and expertise are valuable, and I want to express my gratitude for your willingness to participate in the review process.  If you have any questions or concerns in the future, please do not hesitate to let me know. If you have any other suggestions or comments, please also feel free to share them with me so that I can improve my manuscript.

Thank you again for your support and assistance. I look forward to continuing to work with you.

Best regards,

[Xiaoyuan Jing]

Reviewer 4 Report

This manuscript presents results of a laboratory experimental study that aimed to investigate effects of micro-plastics (MP) on physical properties and hydraulic parameters of farm land soil. 

The experiment was well designated and executed, manuscript is generally well written and subject of investigation is relevant to interests of Agronomy journal, I have therefore no doubts to recommend its publication. The manuscript is generally well written and structured, and I did not find any significant obstacles. In fact, I would say it is virtually almost ready for publication and I have only very few suggestions concerning mostly language and Figures preparation, for instance:

Abstract:

Line 9: “Soil physical properties are the main factors that impact soil fertility …”

The word “impact” implies strong and generally negative effects of something. I would therefore suggest rephrasing of that statement.

Introduction:

Line 39: “… impacting soil health …”

As indicated above, the word “impact” implies strong and generally negative effects of something. I would therefore suggest rephrasing of that statement.

Line 43: “(MP accumulation) … destroy the soil structure integrity … ”

MPs are likely to have negative effect on the soil structure but they do not destroy it. Writing “… impact the soil structure … ” reads better in that case.

Materials and Methods:

Lines 86-87: “Silt loam contains clay, silt, and sand in approximate proportions of 12.58, 72.02, and 15.40%, respectively“.

Literature citation for these data is needed.

Results:

Line 177: “3.1.2. Soil water retention curve” -> “3.1.2. Soil water retention curve (SWRC)”

For clarity, I would suggest adding abbreviation into the section title.

Line 206: “this decrease was lack of …”

Grammar correction needed. “this decrease was lacking of …” would probably read better.

Line 217-219: “this decrease was lack of …”

It is unclear to what the value “(0.01%)” refers to? Grammar correction of that sentence might be necessary.

Page 13, Line 1-3: “this decrease was lack of …”

This text seems to be continuation of a paragraph from the page 11 and should be moved there.

Figure 1.

The meaning of different groups of columns (i.e., T0, T1, T3, T5) and different colors of bars should be explained in the Figure Caption. I would also suggest increasing size of fonts on the figure. Particularly letters indicating significant differences are so small that they are barely readable, but size of other fonts should also be increased.

Figure 2.

Explanation for different symbols and colors in the Figure Caption is necessary. I would also suggest increasing size of fonts given on the figure as they are mostly barely readable.

Figure 3.

See my comments regarding Figure 2.

Figure 4.

See my comments regarding Figure 2.

Figure 5.

See my comments regarding Figure 1 and 2.

Figure 6.

See my comments regarding Figure 1 and 2.

Figure 7.

The Figure Caption should provide explanation (1) of the meaning of values given bellow the horizontal axis and (2) what bar range whiskers represent.

Table 1.

For clarity, all parameters should be clearly defined/explained in the Table Caption.

Discussion

Lines 35-35: “… MP concentration alters the soil permeability by replacing the original soil pores.”

I do not think it is correct to say that MPs replace the soil pores. Pores are not replaced but filled up with MPs.

Lines 83-84: “… MP application alters the soil pore size distribution and porosity”

This reads “… MP application alters the soil pore size distribution and (alters soil pore size) porosity”. Restricting the sentence is needed. The following would read better “… MP application alters the soil porosity and pore size distribution”.

Line 84: “… to increase soil water retention” -> “… increasing soil water retention”

------------------------------------------------------------------------------

Round 2

Reviewer 1 Report

I have provided my remaining comments in uppercase, blue text here.
